# Analysis of the Number and Type of Vaccinations Performed among Polish Soldiers in 2018–2021

**DOI:** 10.3390/ijerph192113724

**Published:** 2022-10-22

**Authors:** Magdalena Zawadzka, Ewelina Ejchman-Pac

**Affiliations:** 1Department of Epidemiology and Public Health Lodz, Medical University of Lodz, 90-419 Lodz, Poland; 2Independent Department of Epidemiology, Military Institute of Hygiene and Epidemiology, 01-163 Warsaw, Poland

**Keywords:** vaccinations, military (soldiers), infectious diseases

## Abstract

Vaccination is a very common topic, but it is rarely raised or discussed with respect to military members. Soldiers are one of the main professional groups to be immunized on a regular basis. The military actively participates in research on new vaccine preparations. This paper presents data from 2018–2021 on vaccination among Polish soldiers. The material obtained from the Central Register of Vaccination for Professional Soldiers was analyzed using descriptive statistical methods. The number of injections performed in a given period depends on the location of the ongoing missions and the vaccination schedule specific to a given Polish Military Contingent. In Poland, soldiers undergo preventive vaccinations in accordance with the scheme developed by the Armed Forces Operational Command, taking into account the specific nature of the service, epidemiological risks and the calendar of current preventive vaccinations. Soldiers serving abroad are immunized against typhoid, hepatitis A, hepatitis B, rabies, measles, tick-borne encephalitis, Japanese encephalitis, polio, diphtheria, meningococcal disease, chickenpox, cholera and yellow fever. Regular vaccinations for soldiers are necessary to minimize the spread of infectious diseases, and they have a beneficial effect upon the effectiveness of military operations.

## 1. Introduction

Prophylactic vaccination is to be understood as administration of a vaccine (biological preparation) against an infectious disease with the aim of stimulating a specific immune response. The main task of vaccines is to protect against the severe course of the disease and its consequences. Currently, several types of vaccines can be distinguished, and the main types include: attenuated vaccines (containing live strains of microorganisms of low virulence); inactivated vaccines (containing dead microorganisms); antigen-based vaccines; and DNA and RNA-based vaccines (containing fragments of the pathogen’s genetic material) [1].

The scope of prophylactic vaccinations is determined by several factors, including the type of activities a person undertakes at work, during which s/he may be exposed to a biological pathogen that may cause an infectious disease; the epidemiological situation; and other individual reasons. However, these factors are all conditional on the existence of a vaccine against the specific infectious disease [2].

Vaccinations at the Ministry of National Defense (RON) are dedicated to professional and non-professional soldiers. They should be administered with consideration toward the specifics of the service and the related tasks, as well as the potential risks that a given position entail. Military personnel are vaccinated in accordance with the Prophylactic Vaccination Program for Professional Soldiers. This program defines the rules at RON for the practice of immunization against infectious diseases. It consists of a prophylactic vaccination schedule for professional soldiers exposed to specific pathogens, as well as vaccinations performed for epidemiological reasons or for so-called supplementary information [3].

Anti-epidemic protection of soldiers delegated to serve outside Poland is an essential element of the combat value of the troops, and the activities undertaken within this framework, such as preventive vaccinations, aim to create safe working and service conditions. The medical procedures applicable before trip include the aforementioned preventive vaccinations and prophylaxis against selected infectious diseases (for example, against malaria), which occur in the area of planned military operations. Vaccination schemes are developed by the Operational Command of the Armed Forces (DORSZ), with consideration of the specificity of the service, epidemiological risks and the calendar of preventive vaccinations. In Poland, there is a data register to record soldiers’ vaccinations (Central Register of Vaccination for Professional Soldiers). The implementation of vaccination is supervised by the Military Sanitary Inspection and the Military Pharmaceutical Inspection, whereas the direct responsibility lies with the commanders [2,4].

The current vaccination scheme, developed by the Operational Command of the Armed Forces for soldiers serving outside Poland is as follows: in the first week, candidates are vaccinated against typhoid, hepatitis A and B, rabies, measles, tick-borne encephalitis and Japanese encephalitis. A week later, a second dose of the anti-rabies vaccine is administered. In the following month, soldiers are vaccinated with a second dose against hepatitis B, typhoid fever, tick-borne encephalitis, and Japanese encephalitis, as well as a third dose of rabies vaccine and a single vaccine against polio and diphtheria. The next step in the vaccination scheme consists of the administration, two months after the first injection, of the meningococcal A, C, Y and W-135; varicella (second dose); and cholera (two doses at two-week intervals) vaccines. After three months, candidates delegated to serve outside Poland receive one dose of vaccine against yellow fever and two doses against meningococcal B (two weeks apart) [3]. 

The purpose of this study is to analyze the number of vaccinations performed among Polish soldiers in 2018–2021.

## 2. Methods

Data on the number of vaccinations performed, mainly among candidates qualified for service (work) outside Poland, were obtained from the Central Register of Vaccination for Professional Soldiers kept in the SEW Military Record System online. The data were made available by the Epidemiological Response Center of the Armed Forces (CRE SZ). The data were statistically analyzed using descriptive methods. 

## 3. Results

Candidates qualified for service (work) abroad are vaccinated with both monovalent and polyvalent preparations. The number of injections performed in a given period is strictly dependent on the ongoing missions and the vaccination schedule specific to a given PMC (Polish Military Contingent).

The number of diphtheria vaccinations is relatively stable; the most injections were given in 2019 (*n* = 2927), and the least were given in 2021 (*n* = 1765). The diphtheria vaccination curve is similar to that of the monovalent polio preparation, which received the highest number of vaccinations in 2019 (*n* = 2855). From the fourth quarter of 2020 to the second quarter of 2021, this vaccine was not administered to soldiers, as a result of the use of the three-in-one vaccine against diphtheria, tetanus and polio. Prevention of typhoid and tetanus was administered with the use of a combined or monovalent preparation. The most common type of vaccine was the polyvalent one, which was applied several times more often than the single preparation (Figure 1 and Figure 2, Table 1).

The four-in-one polyvalent preparation for measles, mumps, rubella and chickenpox was used sporadically. In the analyzed period, a very large increase in the number of vaccinations against measles, mumps and rubella can be observed in the period from Q4 2018–Q1 2019, while in the remaining months, the number of vaccinations fluctuates at a relatively constant level, i.e., between 100–200 administered preparations per month (Figure 3). The highest number of vaccinations against measles, mumps and rubella was performed in 2018, and against chickenpox in the following year, i.e., *n* = 1917 and *n* = 1995, respectively (Table 1).

Soldiers serving abroad are vaccinated against rabies, tick-borne encephalitis (TBE) and meningococcal disease. The number of injections performed for the above-mentioned vaccinations varies; however, there is a noticeable similarity in the curve distribution (Figure 4). The number of injections against tick-borne encephalitis was, on average, almost two times higher than those against rabies or meningococcal disease in 2018. The highest number of administered preparations for TBE was given in 2018 (*n* = 13,321), while against rabies and meningococcal disease, the highest number was given in 2020, i.e., *n* = 8714 and *n* = 7854, respectively (Table 1).

With respect to hepatitis A and B, monovalent vaccine preparations are used twice as often as combined preparations. The number of vaccinations against viral hepatitis A and B is relatively constant, except for an increase in June–July 2020 and June–July 2021 of the vaccinations against hepatitis B, and in December–January 2021 against hepatitis A (Figure 5). The highest number of vaccinations for viral hepatitis B occurred in 2021 (*n* = 3589); however, this figure is comparable to 2020 (*n* = 3484). Regarding hepatitis A and hepatitis A + B, the highest number of vaccinations both occurred in 2019, i.e., *n* = 3808 and *n* = 2493, respectively (Table 1).

There were three increases in the vaccinations against cholera, which reached their peaks at the turn of 2018/2019, September–October 2019 and June 2019. The number of vaccinations against Japanese encephalitis is more than two times lower than against cholera and three times greater than against yellow fever (Figure 6). The number of vaccinations performed was the highest (*n* = 7720) for cholera in 2021, and for yellow fever (*n* = 1348) and Japanese encephalitis (*n* = 4142) in 2019 (Table 1).

## 4. Discussion

The largest armies in the world can boast a long-standing tradition of research on vaccination, which is due to the fact that the availability and health of soldiers contribute to the effectiveness of military operations. The American army has the most significant achievements. Schmaljohn et al. reports that between 1945–1995, the US military co-financed studies on 10 different vaccines (against, among others, meningococcus, hepatitis A and B, influenza, etc.), which are also used by civilians [5].

The poorly hygienic conditions of soldiers are conducive to the spread of infectious diseases. These diseases account for even more deaths in the military than the injuries suffered in combat. Long-term data show that the implementation of the vaccination program for soldiers not only provides health benefits, but also financial benefits, as it inhibits the spread of infectious diseases and reduces the costs associated with the treatment of serious complications within the army [2,3].

Mura et al. published a paper concerning the vaccination program of the French army. This program was created as a result of many years of observations of infections on battlefronts. The program is being established on an ongoing basis by the Technical Committee for Immunization in conjunction with the guidelines of the French Ministry of Health. The recommendations of the Ministry strictly depend on the extent to which infectious diseases are a problem in the country, on the location in the country to which military units are sent and on vaccine availability. Compulsory vaccinations that soldiers are subjected to result from: being in a large group of people, sustained injuries (possible wound infections) and poor sanitary conditions (diseases of the digestive system), among other things. The authors analyzed the data from 2017, which show that the most vaccinations were performed against influenza (103,502) and meningococcus (66,452) at that time [6]. Additionally, Mimouni et al. proved in their publication that since vaccination against meningococcal disease was introduced in the Israeli army, the number of infections has decreased [7]. In turn, the analysis presented in our article shows that the most vaccinations were performed against TBE, with more than twice as many vaccinations as there were against meningococcal disease.

Jae Hyoung et al. presented data on Korean soldiers, which showed that the frequency of clinically confirmed cases of hepatitis A is statistically lower in the vaccinated group compared to the unvaccinated group (0.5/100,000 in the vaccinated group, 2.06/100,000 in the unvaccinated group). The analysis was carried out in 2013–2016, after the obligatory immunization of soldiers was introduced [8]. In Poland, vaccination against hepatitis A and B is also mandatory. As demonstrated in our study, the frequency of monovalent vaccination against hepatitis A and B is more than twice as high as that of the combined preparation.

When addressing the subject of vaccination among soldiers, it is worth analyzing the effectiveness of the available vaccines, because only proven preparations provide some protection against infection. Ferlito et al. published data on the presence of antibodies against measles, mumps and rubella; hepatitis A; polio; and influenza among Italian soldiers. The authors observed that antibodies against measles, mumps and rubella are at levels that ensure protection. After 9 months, that number increased even more, which proves an effective immune response. In the case of polio and HAV, the results were similar (an increase in the level of antibodies after the administration of the vaccine, compared to the baseline) [9]. According to the legal regulations in Poland, candidates delegated to serve abroad are subjected to a single vaccination against polio; the number of vaccinations in the analyzed period was variable. The lowest number of vaccinations was performed in 2019, which constitutes a value three times lower than in the previous year when the number was the highest in the analyzed period.

Population immunity is an important factor inhibiting the spread of infectious diseases. In their work, Lewis et al. determined the level of antibodies against measles, mumps and rubella among newly recruited members of the US Air Force (between April 2013 and April 2014). Using serological methods, 32,502 samples were tested, 22,878 of which were found to have antibodies against measles and rubella. In addition, 20,064 people tested positive for anti-mumps antibodies. Unfortunately, over 38% of the surveyed soldiers tested negative for at least one of the three analyzed types of antibodies, while nearly 4% were seronegative for all three types. The results were lower than expected (compared to the studies carried out on civilians). According to the authors, the titer of antibodies is not sufficient to achieve herd immunity, which may result in an increase in the number of people infected with the diseases analyzed in the article. Scientists suggest that authorities should monitor the situation and take appropriate action [10]. In Poland, appropriate preventive measures, consisting of obligatory vaccinations of soldiers against mumps, measles and rubella, were carried out. Every year, 1.5 thousand candidates delegated to serve abroad are vaccinated.

Japanese encephalitis (JE), which is caused by a virus of the genus Flaviviridae, is the most common cause of encephalitis in Asia. Tun et al. tested a group of 195 people in order to study the effectiveness of the vaccine against JE. The antibody titer was determined using the immunocytochemical method (ELISA). IgG 6 months after vaccination, which was high after the first dose, which proves the effectiveness of the preparation against JE. Its effectiveness varies depending on the population and ranges between 85%–98%, but its level is high enough to minimize the risk of endemic outbreaks of JE epidemics [11]. Preventive measures are also taken in the Polish army. As demonstrated in the study, the number of vaccinations against JE is three times greater than that against yellow fever and, at the same time, two times lower than that against cholera.

Yellow fever (YF) is an infectious viral disease transmitted by mosquitoes, mainly found in tropical areas. The course of the disease can vary from mild to debilitating, or even fatal. Despite the vaccine’s availability, a large group of people (mainly those living in the equatorial belt) remain unvaccinated. It is therefore necessary to administer the vaccine to members of military units deployed in areas with risk of yellow fever virus transmission. As part of these activities, vaccinations are also performed in the Polish army when it is necessary for them to go to such areas at risk of yellow fever. In the analyzed period of the study, the average number of YF vaccinations performed fluctuated around 900 vaccines per year. In addition, Hall et al. analyzed the efficacy and safety of the anti-YF preparation in a group of pregnant US soldiers. The vaccine was given to 195,455 women, and then data on the effects of exposure were collected (including problems with conception, premature delivery, low birth weight, birth defects, etc.). The obtained results indicate no harmful effect on the course of pregnancy and the health of newborns in comparison to the unvaccinated population [12].

Another dangerous disease to which soldiers are exposed is cholera. In their publication, Khatib et al. presented data on the effectiveness of vaccination with an oral preparation among the inhabitants of Zanzibar, after an outbreak of the disease. The effectiveness of the vaccine was established based on the number of cholera cases detected and was estimated at 79% (in the group that received two doses) and 46% (in the group that received one dose). Additionally, it has been observed that the probability of contracting cholera for unvaccinated people is lower than in a population with a high vaccination rate [13]. According to the present study, Polish soldiers leaving for missions abroad are vaccinated against cholera in two doses; on average, over 5000 vaccinations with the abovementioned preparation are performed yearly.

When discussing the topic of soldiers’ vaccinations, attention should be also paid to abdominal dura. In their publication, Porter et al. examined data from 1998–2011, comparing two available preparations (oral and intramuscular) administered to members of the US Army. Both preparations are safe, because no statistically significant adverse reactions to vaccination or allergic reactions were noted. The military received the intramuscular vaccine much more often. Over 13 years of analysis, an increase in the number of vaccinations against typhoid fever has been reported, which results from the increase in military activities carried out by the US military [14]. The basic vaccination scheme of candidates qualified for service outside the territory of the Republic of Poland also includes compulsory immunization for typhoid fever. As presented above, soldiers are administered a combined preparation for typhoid fever and tetanus.

To sum up, vaccination of soldiers against common infectious diseases is necessary. The references we used confirm the effectiveness and safety of the preparations available on the market. It is also worth noting that compulsory vaccinations are necessary to obtain population immunity and inhibit the spread of infectious diseases.

## 5. Conclusions

There is a need to monitor the vaccinations carried out among soldiers. Regular vaccinations of soldiers are necessary to minimize the spread of infectious diseases and have a beneficial effect on the effectiveness of military operations.

## Figures and Tables

**Figure 1 ijerph-19-13724-f001:**
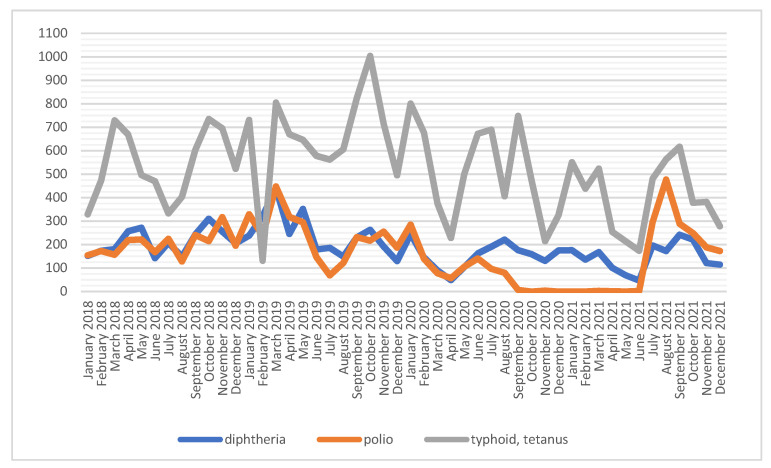
Number of vaccinations against diphtheria, polio, typhoid fever and tetanus among soldiers in 2018–2021.

**Figure 2 ijerph-19-13724-f002:**
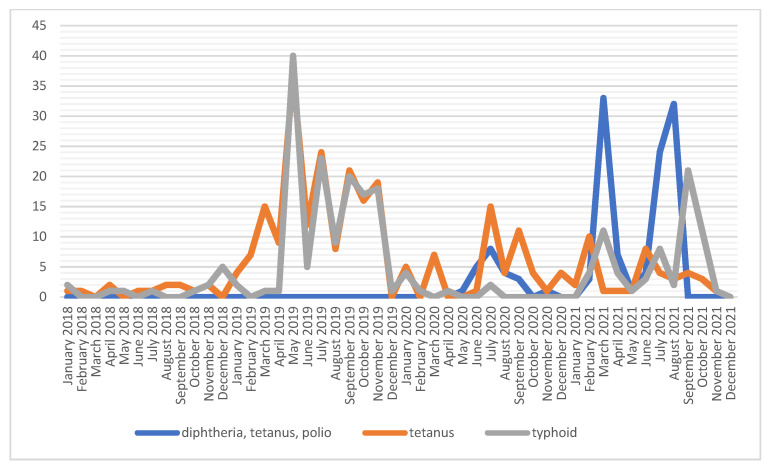
Number of vaccinations against diphtheria, tetanus, polio and typhoid fever among soldiers in 2018–2021.

**Figure 3 ijerph-19-13724-f003:**
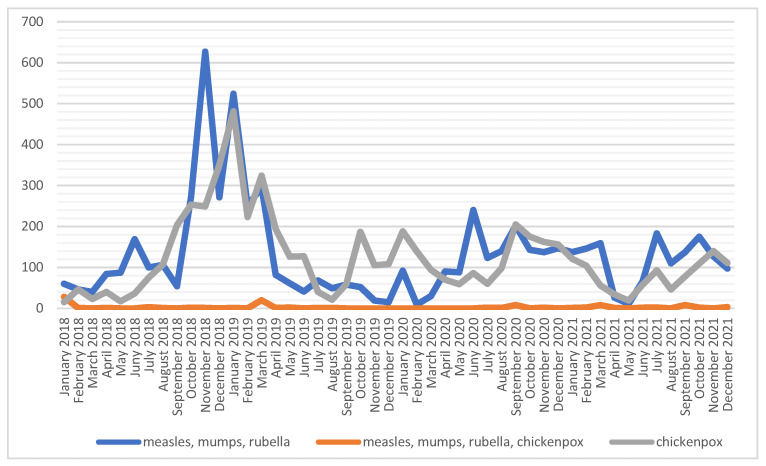
Number of vaccinations against measles, mumps, rubella and chickenpox among soldiers in 2018–2021.

**Figure 4 ijerph-19-13724-f004:**
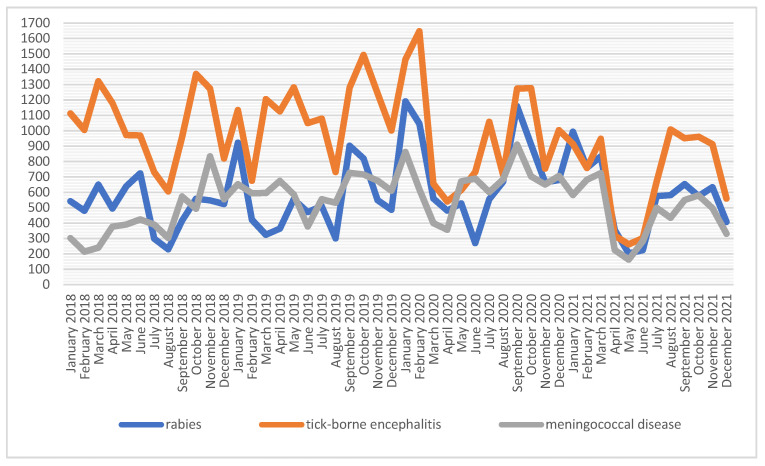
Number of vaccinations against rabies, tick-borne encephalitis and meningococcal disease among soldiers in 2018–2021.

**Figure 5 ijerph-19-13724-f005:**
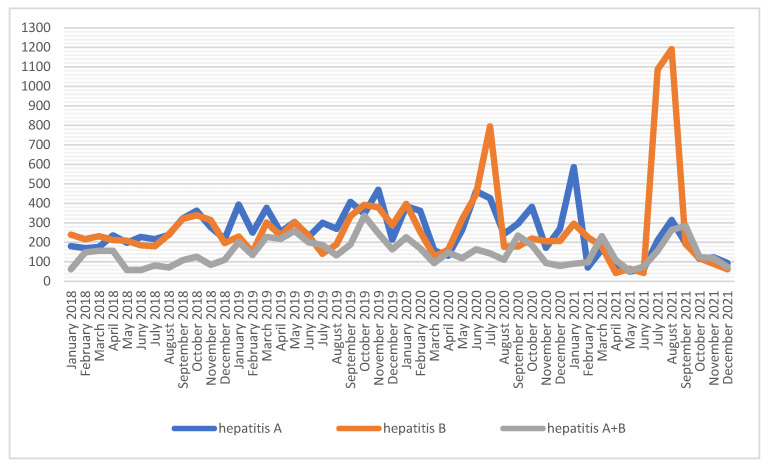
Number of vaccinations against hepatitis A and B among soldiers in 2018–2021.

**Figure 6 ijerph-19-13724-f006:**
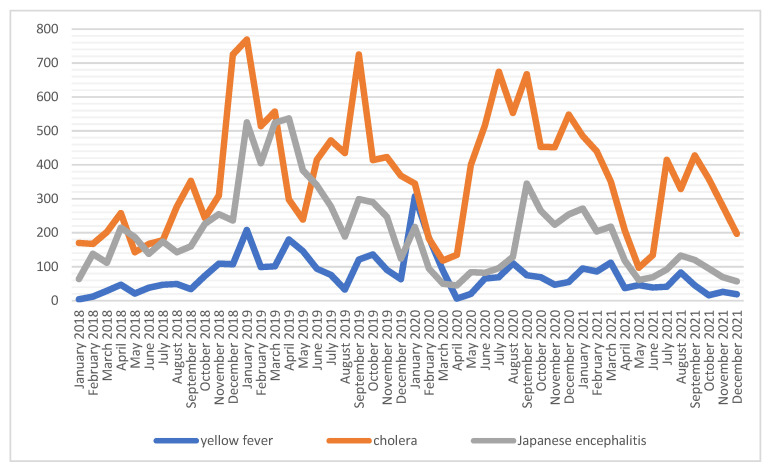
Number of vaccinations against yellow fever, cholera and Japanese encephalitis among soldiers in 2018–2021.

**Table 1 ijerph-19-13724-t001:** Number of vaccinations performed among soldiers in 2018–2021.

Vaccine	Year			
	2018	2019	2020	2021
diphtheria	2548	2927	1848	1765
polio	2415	2857	995	1682
diphtheria, tetanus, polio	0	0	29	104
typhoid, tetanus	6458	7765	6112	4854
tetanus	13	172	52	38
typhoid	13	137	8	66
measles, mumps, rubella	1917	1515	1441	1374
chickenpox	1414	1995	1492	969
measles, mumps, rubella, chickenpox	37	26	13	29
rabies	6104	6628	8714	6796
tick-borne encephalitis	13,321	13,301	11,733	8564
meningococcal disease	5099	7297	7854	5552
hepatitis A + B	1217	2493	1760	1673
hepatitis B	2879	3174	3484	3589
hepatitis A	2819	3808	3543	2069
cholera	3194	5629	5044	7720
Japanese encephalitis	2048	4142	1886	1507
yellow fever	570	1348	1098	645

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
