# Peer review of "Analysis of the Number and Type of Vaccinations Performed among Polish Soldiers in 2018–2021"

_ijerph, 2022, doi:10.3390/ijerph192113724_

Round 1

Reviewer 1 Report

Analysis of the Number and Type of Vaccinations Performed Among Polish Soldiers 2018-2021

Abstract

(1)   4th line down should add an s to vaccination to read vaccinations

Introduction

(2)   What is meant by so-called supplementary information?

(3)   The purpose statement should be at the end of the Introduction not in the Methods section

Materials and Methods

(4)   The heading should just be Methods.  What are materials?  Start this section with Data on the number of vaccinations…….   not The material consists of data……

(5)   Line 64: What is empirical  material?  Do you mean data was analyzed using descriptive statistics

Results

(6)   In Table 1 change the heading Name to Vaccination.

(7)   Line 74:  Delete In the analyzed period and start the sentence with The number of diphtheria vaccinations……

(8)   Line 100:  Change Candidates for service abroad are subjected to the prevention of rabies to Soldiers serving abroad will be vaccinated against rabies, tick-borne encephalitis (TBE) and meningococcal disease.

(9)   Line 111: Replace In the prophylaxis of hepatitis A…. with     With respect to hepatitis A and B, monovalent vaccine preparations are used twice as often a combined preparations.

(10)  Lind 120: Eliminate the first sentence and start the paragraph with There are three increases in the (eliminate word applied)  vaccinations against cholera….

Discussion

(11)    I suggest the first paragraph in the Discussion section be moved to the Introduction.  It is out of place in the Discussion

(12)      Line 150: replace development with spread ( of infectious diseases)

(13)    What doers epidemiological problems (line 156) mean?  Do you mean “infections”

(14)  Similar to (7) The statement: The recommendations of the Ministry strictly depend on, among others on the epidemiological situation of the country……..    What does epidemiological situation mean?  As I read the sentence the sentence could read: The recommendations of the Ministry strictly depend on the extent to which infectious diseases are a problem in the country and/or location in the country to which military units are sent and vaccine availability.  

(15)  Line 163 – What does vaccine preparations on the market mean? I think you meant to say available vaccines not vaccine preparations on the market

(16)  Line 170:  What article below?  Are you referring to the Hyoung article?  If so, that sentence can be eliminated.

(17)  Line 254 – The examples don’t prove  anything.  Use support instead of prove,

Author Response

Dear Mr/Mrs

Thank  you very much for Your positive review, dedicated time and remarks. All Your suggestions were plased in our manuscript.

Your sincerely

Reviewer 2 Report

I appreciate the authors for their work on analyze the number of vaccinations performed 65 among Polish soldiers in 2018-2021. The result analysis should be discuss in more detail. 

Author Response

Dear Mr/Mrs

Thank you for Your positive review of the manuscript.
Data analysis was prepered on the basis of the available information. The original idea was to present the results in comparison with the total number of soldiers in Poland, but unfortunately the consent of the superior organs was not obtained, because it can weaken security of Poland.

Your sincerely

Reviewer 3 Report

The authors give an update on the number and type of vaccinations performed among Polish soldiers ranging from 2018 to 2021. The introduction has the proper information. The results are clearly represented and the discussion is well done. I recommend the publication of this relevant article as is. 

Author Response

Dear Mr/Mrs

Than  you very much for Your positive review and dedicated time

Your sincerely